# Enabling the Internet of Mobile Crowdsourcing Health Things: A Mobile Fog Computing, Blockchain and IoT Based Continuous Glucose Monitoring System for Diabetes Mellitus Research and Care

**DOI:** 10.3390/s19153319

**Published:** 2019-07-28

**Authors:** Tiago M. Fernández-Caramés, Iván Froiz-Míguez, Oscar Blanco-Novoa, Paula Fraga-Lamas

**Affiliations:** Department of Computer Engineering, Faculty of Computer Science, Centro de investigación CITIC, Universidade da Coruña, 15071 A Coruña, Spain

**Keywords:** diabetes, glucose monitoring, CGM, fog computing, IoT, blockchain, crowdsourcing mHealth, public health, decision support, personalized medicine

## Abstract

Diabetes patients suffer from abnormal blood glucose levels, which can cause diverse health disorders that affect their kidneys, heart and vision. Due to these conditions, diabetes patients have traditionally checked blood glucose levels through Self-Monitoring of Blood Glucose (SMBG) techniques, like pricking their fingers multiple times per day. Such techniques involve a number of drawbacks that can be solved by using a device called Continuous Glucose Monitor (CGM), which can measure blood glucose levels continuously throughout the day without having to prick the patient when carrying out every measurement. This article details the design and implementation of a system that enhances commercial CGMs by adding Internet of Things (IoT) capabilities to them that allow for monitoring patients remotely and, thus, warning them about potentially dangerous situations. The proposed system makes use of smartphones to collect blood glucose values from CGMs and then sends them either to a remote cloud or to distributed fog computing nodes. Moreover, in order to exchange reliable, trustworthy and cybersecure data with medical scientists, doctors and caretakers, the system includes the deployment of a decentralized storage system that receives, processes and stores the collected data. Furthermore, in order to motivate users to add new data to the system, an incentive system based on a digital cryptocurrency named GlucoCoin was devised. Such a system makes use of a blockchain that is able to execute smart contracts in order to automate CGM sensor purchases or to reward the users that contribute to the system by providing their own data. Thanks to all the previously mentioned technologies, the proposed system enables patient data crowdsourcing and the development of novel mobile health (mHealth) applications for diagnosing, monitoring, studying and taking public health actions that can help to advance in the control of the disease and raise global awareness on the increasing prevalence of diabetes.

## 1. Introduction

Diabetes Mellitus (DM), which is usually referred to as Diabetes, is a worldwide chronic metabolic disorder that involves abnormal blood glucose level oscillations that lead to both macrovascular alterations, which affect large blood vessels (coronary arteries, the aorta, and arteries in the brain and in the limbs) and microvascular complications, which affect the kidneys (nephropathy), nerves (neuropathy), and eyes (retinopathy). According to the World Health Organization (WHO) [1], there are three main types of DM:
Type-1 DM (DM1): it is an autoimmune process that leads to remarkably diminished insulin levels.Type-2 DM (DM2): it arises from the compromised function of insulin-producing cells in the pancreas and insulin resistance in the peripheral tissues.Gestational Diabetes Mellitus (GDM): it occurs on pregnant women that suffer from above-normal blood glucose levels.

Due to the health issues related to DM, it is important to monitor vulnerable groups, especially children, the elderly and pregnant women. Such a monitoring has been traditionally performed by taking blood samples through Self-Monitoring of Blood Glucose (SMBG) techniques [2], which have a number of drawbacks (e.g., active involvement of the patient or his/her caretakers, infections) that can be tackled by Continuous Glucose Monitors (CGMs), which are based on a small device with a sensor that takes blood glucose readings 24 h a day [3]. Such measurements make it easier for DM patients to have more precise control over blood glucose, which allows patients the ability to make informed therapeutic decisions. Thus, CGMs may warn patients about hyperglycemia (high blood glucose level) and hypoglycemia (low blood glucose level) in order to take the appropriate preventive measures. Nonetheless, it is worth pointing out that the use of CGMs carry some inconveniences: CGMs are usually expensive (although some countries are starting to subsidize its acquisition), they read glucose concentration values with a delay between 5 and 10 min [4], some CGMs need to be calibrated several times a day by finger-pricking and their typical lifespan is short (it usually goes between 3 days and a couple of weeks, but recent CGMs keep on working up to six months).

Despite CGM current drawbacks, the concept of CGM opens the possibility of creating Internet of Things (IoT) devices that provide rapid warnings and are able to make autonomous decisions when actions must be performed as fast as possible to avoid dire outcomes [5]. Thus, an IoT CGM can make use of a remote cloud system where information is stored and where rule-based decisions can be taken (e.g., to warn a doctor when the patient’s blood glucose level is above or below a specific threshold). However, traditional cloud computing architectures have certain limitations: all the information and decisions are centralized and managed, in general, by a third party; the cloud availability may be compromised by overloading or by cyber-attacks; and, due to physical distance, there may be a lag between the cloud and the patient resulting in a delay between the decision to perform an action and the communication to the patient to do so, which may be too long in some cases. Fortunately, for such scenarios where a fast response and low communications overhead are required, other paradigms have been successful by moving computing capabilities from the cloud towards the edge of the network [6]. One of such paradigms is fog computing, which transfers the cloud computational and communication capabilities close to the sensor nodes in order to minimize latency, to distribute computational and storage resources, to enhance mobility and location awareness, and to ease network scalability while providing connectivity among devices in different physical environments [7,8].

Security and trustworthiness are other problems that arise when collecting and processing the data sensed by IoT devices. Regarding security, some authors proposed energy-efficient mechanisms to make use of high-security cipher suites in IoT devices, since they are usually constrained in terms of computational resources, especially when they rely on batteries [9]. With respect to trustworthiness, it is essential when data are shared with third parties. Therefore, the collected data (e.g., for doctors or for autonomous systems that base their decision-making on the received blood glucose values) should be validated.

Doctors and researchers also need data in order to improve the existing knowledge on DM and to look for a potential cure. Such medical data are usually difficult to obtain due to different reasons (e.g., lack of access to useful data, existing laws, lack of user trust), so it is important to study new ways to automate data collection on a large scale. Crowdsourcing is one potential technique, since it is able to make use of the collective intelligence of an online community to research and to develop innovative human-centered approaches and novel products and services [10]. However, there are not many public health crowdsourcing applications [11], specially in the area of mobile health (mHealth), which harnesses smartphones and wireless communications technologies to ease the access to healthcare solutions. There are even less decentralized mobile crowdsourcing healthcare applications, since such systems usually depend on a public health central authority or on a private company, which may be a single point of failure prone to cyber-attacks.

To tackle the previously mentioned issues, this article details the design and implementation of a system that includes the following contributions, which have not been found together in the previous literature:
A description of an IoT CGM-based system able to monitor glucose concentration values remotely and quickly inform patients and/or their care givers of a dangerous situation.The design of a system able to provide fast warnings by using fog computing gateways when the user is located in specific scenarios (e.g., inside the patient’s home, in a hospital or in a nursing home).A description of a mHealth blockchain-based decentralized architecture that does not depend on any central authority.The proposal of a data crowdsourcing mechanism that encourages collaboration by rewarding patients.The evaluation of the proposed architecture to assess the performance of the described decentralized storage and blockchain.

The rest of this article is structured as follows. Section 2 reviews the previous work on CGM applications and on the use of fog computing, blockchain and crowdsourcing for healthcare applications. Section 3 details the proposed communications architecture and the designed crowdsourcing incentive mechanism, while Section 4 describes their implementation. Finally, Section 5 is devoted to the experiments and Section 7 to the conclusions.

## 2. Related Work

### 2.1. CGM Applications

During the last decade different authors studied the use of CGMs in order to apply them in different scenarios. For instance, in [12] the possibility of adding intelligence to CGM sensors is analyzed in order to issue alerts when glucose levels are not within the appropriate range (e.g., in case of a hypoglycaemia/hyperglycaemia). The same authors focused later on reviewing the state of the art on contributions related to the development of hardware and smart algorithms for CGMs [13]. Similar reviews on the evolution of CGM technology can be found in [14,15,16].

Other CGM features have been studied by other authors. For instance, some authors focused on analyzing off-the-shelf CGM calibration algorithms and on how the amount of glucose in blood plasma is obtained [17]. Other researchers studied how CGMs detect losses in the performance on insulin infusion set actuation [18], which may derive into prolonged hyperglycemia in Type-1 DM patients. A CGM is also able to decipher the influence of daily habits on the health of DM patients. For example, physical exercise clearly alters glucose concentration levels on Type-1 DM patients: glucose regulation problems may arise during or after exercising and even when performing certain daily activities [19].

Finally, it is worth mentioning that a CGM can collaborate with an artificial pancreas in order to provide the optimal insulin dose through an insulin pump. However, there are different parameters that influence the infusion of insulin, with physical activity being one of the most challenging as it may result in hypoglycemia in Type-1 DM patients [20].

### 2.2. Fog Computing, Blockchain and Crowdsourcing for Healthcare Applications

In the last years, cloud computing has achieved remarkable success thanks to its ability to offload computational-intensive tasks from clients [21]. Nonetheless, in applications where low latency responses are needed, other paradigms like fog computing have proven to be valuable [6]. Fog computing is usually considered a paradigm that is an extension of cloud computing where part of the computational and communication capabilities of the cloud are moved close to the sensor nodes [7], which derives into several advantages [22]:
Novel IoT healthcare real-time applications can be provided thanks to decreasing latency.The fog allows for sharing computational and storage resources, which can be harnessed by distributed wireless sensor networks that can be deployed in hospitals or in other healthcare facilities.The fog can also connect multiple physical environments that are far from each other, easing device and user interaction and thus facilitating the development of new potential healthcare services.Since the fog can be scaled easily, it provides flexibility and ease of growth to large deployments.

Despite the mentioned benefits and its multiple applications in diverse fields [23], there are not many practical examples of the use of fog computing in healthcare applications. A good compilation of fog computing based healthcare applications can be found in [24], which describes potential applications for monitoring Chronic Obstructive Pulmonary Disease (COPD) patients [25], tools for in-home Parkinson disease treatments [26] or hospital platforms that make use of e-textiles [27] and wireless sensor networks [28]. Similarly, other researchers proposed applications for healthcare monitoring in smart homes [29] or for diagnosing and preventing outbreaks of certain viruses [30].

Another challenge faced by today’s healthcare researchers and practitioners is the lack of interoperability between different technological platforms. Incompatible, unscalable and independent systems hinder the development of novel end-to-end patient-centered research and healthcare solutions. Unfortunately, it is difficult to exchange information among healthcare providers, patients and third-parties (e.g., insurance companies, governments, app developers). Records are often incomplete, fragmented or unavailable at the point of care and it is difficult to access patient’s health information [31,32]. The integration complexity lies mainly in the lack of access outside a specific healthcare environment and in the use of incompatible and proprietary software and hardware. Although there are some standards (e.g., Health Level Seven (HL7) Fast Healthcare Interoperability Resources (FHIR), HL7 Clinical Document Architecture (CDA), ISO13606, openEHR, CDISC Operational Data Model (ODM)) for exchanging data between trusted parties, their implementation requires data mapping and additional interface adaptations [32].

A promising alternative to solve the aforementioned problems consists in using Distributed Ledger Technologies (DLTs). Examples of DLT platforms currently in use are Ethereum [33], Hyperledger Fabric [34] or IOTA [35]. DLTs, specifically blockchain, are predicted to be key technologies within the Industry 4.0 era, since they guarantee the exchange of information between different stakeholders and interested parties that do not necessarily trust each other [36]. Moreover, blockchain holds the promise of enhanced data transparency, trustworthiness, immutability, privacy and security. Furthermore, it enables Peer-to-Peer (P2P) transactions, decentralized Apps (DApps), operational efficiency and a high degree of automation thanks to the use of smart contracts that execute code autonomously [37].

There are recent reviews on the usage of blockchain to enhance the healthcare sector [31,32,38,39,40,41,42]. Such works essentially analyze promising technologies, potential applications and discuss potential challenges for their further adoption. Currently, in the literature there are only a couple of recent preliminary works that use a blockchain for diabetes research and care. An example is [43], where the authors give an overview of a blockchain-based architecture that implements data and access management. An implementation of an Ethereum IoT platform architecture to take care and monitor DM patients can be found in [44].

Finally, it is worth noting that, with the rise of IoT [45,46], crowdsourcing is gaining momentum in a wide range of sectors and tasks [47,48,49,50,51]. In healthcare, crowdsourcing has been mainly employed to accomplish problem solving, data processing, surveillance/monitoring, and surveying [52], but there are not many mobile crowdsourcing applications [11,53] and even less decentralized while focused on healthcare [48]. In health crowdsourcing, engagement is essential since it can transform users from mere passive recipients of information to active participants in a collaborative community, raising awareness for diseases like diabetes and helping to improve their own health as well as the health of those around them [54]. In addition, there are incentive mechanisms that enable community participation [48]. For example, the authors of [55] proposed an incentive mechanism to encourage hospitals to share high-quality data, which can then be aggregated to generate prediction models with higher accuracy rates.

## 3. Design of the System

### 3.1. Communications Architecture

The communications architecture proposed for the solution detailed in this article is illustrated in Figure 1 (on the left). As it can be observed at the bottom of the Figure, CGMs are responsible for collecting glucose concentration values from remote patients. Such values are then read by a smartphone, which periodically scans the available wireless networks looking for neighboring fog computing gateways. Thus, the smartphone may be in one of the two following scenarios:
In an area covered by a fog computing gateway. In this scenario the smartphone joins the network created by the fog gateway and sends to it the data collected from the CGM sensor. This may be the case of homes and certain buildings (e.g., hospitals, nursing homes), where fog gateway infrastructure can be easily deployed and managed.Out of range of the deployed fog computing gateways. Patients may move to certain areas where fog gateways are not in range (e.g., outdoors, in certain areas of a home/building). In such scenarios the smartphone should detect the lack of fog gateway connectivity and send the collected data directly to remote servers or services on the Internet.

In the case of making use of a fog gateway, an mHealth fog service will be run on it. Such a service receives the sensor data from the smartphone and decides whether it is necessary to take a specific action. For instance, if the mHealth fog service detects a down trend on the glucose levels that may lead to hypoglycemia, it can warn the user or his/her caretakers (note that in Figure 1 it is considered the existence of users like relatives, doctors or nurses that may be near the patient (local users) or in remote places (remote users), which may be responsible for looking after the patient).

Besides providing fast responses to users, the mHealth fog service is also able to send the collected information to a remote server, to a decentralized storage system or to a decentralized ledger:Remote server: it is essentially a front-end that provides a web interface to remote users in order to allow them to access the stored information in a user-friendly way. This server also runs a back-end service that is responsible for sending notifications to remote users through SMS or instant messaging services.Decentralized storage system: it is able to provide redundancy, cyber-attack protection and DApp support to the stored information. This storage system is able to replicate the collected information and distribute it automatically among multiple nodes. In this way, although one storage node is not available (e.g., due to maintenance, to a malfunction or to a cyber-attack), the information can be accessed through other nodes.Decentralized ledger: it is included to provide transparency and decentralization, and to enhance data authenticity and security. Specifically, a blockchain can be used to preserve user anonymity and transaction privacy, so that patient data can remain private to non-authorized parties [56]. Moreover, transparency is provided to the authorized third-parties that access the stored information, which allows them to monitor and analyze it in order to determine whether it has been tampered [57]. Furthermore, a blockchain provides decentralization by distributing the data among peers and thus avoiding the involvement of middlemen [58]. Regarding data authenticity, a blockchain can be used to provide accountability, which is essential to trust the information collected from the patients. With respect to smart contracts [37], they are able to automate event triggering by monitoring when certain requirements are fulfilled. For instance, a smart contract may be used together with an oracle (i.e., an external agent that retrieves and validates certain real-world information that is later sent to a blockchain) to detect that a user is going to run out of sensors and then automate the purchase of new ones. The designed decentralized ledger also includes a cryptocurrency-based incentive mechanism (whose functioning is detailed in the next subsection) to motivate patients to share their data in the crowdsourcing platform.

### 3.2. Data Crowdsourcing Incentive Mechanism

One of the objectives of the proposed system is to create the basis for a mHealth data crowdsourcing ecosystem that can be harnessed by third parties in order to improve the existing knowledge and research on DM. To achieve such an objective, it is essential to motivate the patients to upload their CGM sensor data to the decentralized storage nodes so that anyone, once authorized, can access them.

The first incentive measure to ease this data collection process is its automation, which is carried out by the devised CGM-based IoT system. Thus, after setting up the system (i.e., after installing the required fog gateways and the smartphone app), the user only has to check his/her glucose levels. The rest of the processes (e.g., data processing/uploading/distribution, notifications) do not require the user intervention.

The second incentive measure is related to patients that may not be willing to share their data publicly or for free through the proposed crowdsourcing platform. In such a case, the blockchain allows for implementing a private system where patients receive some sort of reward when they contribute with their mHealth data, so the more data they add to the system, the higher the reward.

The reward system is based on the use of a metacoin, which is a digital cryptocurrency that uses an already-deployed blockchain, but which implements additional transaction logic over such a blockchain. For this article, a metacoin called GlucoCoin (
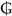
) was created on the Ethereum blockchain. GlucoCoin requires creating a digital wallet for every participant and implements all the software operations needed by a cryptocurrency, which essentially involve getting the balance from a user and sending an amount from one user to another.

GlucoCoin can be used not only for rewarding the users for their data, but it can be used by them to purchase items or services. For instance, as it is later described in Section 4.2, patients may use their GlucoCoins to automate the purchase of new CGM sensors when they are going to expire.

## 4. Implementation

### 4.1. Implemented Architecture

In order to implement the architecture depicted on the left of Figure 1, the components indicated in Figure 1 on the right were selected. As it can be observed at the bottom of the Figure, Android smartphones are used to read the Abbot’s Freestyle Libre CGM sensor. If the patient is indoors, in an area where there is fog computing gateway deployed, the smartphone can communicate with it. Such a gateway is essentially a Single-Board Computer (SBC) that runs the required software. The smartphone can also send the collected values to other decentralized nodes on the Internet or to the Ethereum blockchain, which can be accessed by remote users. In addition, such users can access the data through a front-end that shows a user-friendly web interface. The following subsections describe in detail the characteristics, software and regular functioning of every component.

#### 4.1.1. CGM Sensor

There are a number of companies that manufacture and distribute commercial CGMs, being Dexcom [59], Medtronic [60] and Abbot [61], the most active in the field. Their most relevant and latest CGMs are compared in Table 1 together with some popular sensor-integrated insulin pump systems that are connected to specific CGM systems (e.g., MiniMed 670G). Additional systems and their main characteristics can be found in [62].

Among the analyzed CGMs, Abbott’s Freestyle Libre was chosen. Although its reading range is limited (up to 10 cm) due to the use of Near Field Communication (NFC), the sensor lasts longer than others provided by Dexcom or Medtronic (14 days versus roughly a week), it is cheaper than other solutions (e.g., a kit with one reader and sensors currently costs € 169.69), finger-pricking is not necessary on a daily basis to calibrate the sensor and it is more accurate than other systems (i.e., it has a lower Mean Absolute Relative Difference (MARD)).

Note that it would be possible to make use of the second generation of Abbot’s CGM (Freestyle Libre 2), which provides better MARD, but it was not available for purchase at the beginning of the work related to this article and, in practice, it is essentially the same device.

Abbott’s Freestyle Libre sensor is placed on an arm by using a single-use applicator (in Figure 2, on the left). On the right of Figure 2 it is shown the sensor adhered to the optimum location, in the back of the upper arm. After activating the sensor, it takes roughly one hour before the first readings are available. Then, according to Abbot, the sensor will collect data during a maximum of 14 days (in practice, it has been observed that the sensor keeps on collecting data during almost one day more). In addition, note that each Freestyle Libre sensor also has a commercial expiration date that limits its usage time interval.

To collect readings from the sensor, the user just has to scan it with Abbot’s reader or with an NFC-enabled smartphone that runs Abbot’s LibreLink app. The sensor collects readings automatically every 15 min. It is necessary to scan the sensor at least once every 8 h, since the sensor stores up to 8 h of readings. Readings are stored in the internal memory of the sensor, whose structure was previously reverse engineered (at least in part) by open-source initiatives like LibreMonitor [63] or LimiTTer [64]. The most relevant parts of the sensor internal memory are illustrated in Figure 3. As it can be observed, the memory is structured into 344 bytes that are divided into a header, a footer and the main memory fields, which are the following:
Pointer fields: bytes 27 and 28 store, respectively, the position where the last trend and historic values were written.Trend value fields: they are 16 fields of 6 bytes between byte 29 and byte 124 that store the latest trend measurements. Such trend measurements are obtained every minute for the last 16 min. In the position that is indicated by the Trend Pointer it is placed the current measurement, while the average of the other 15 measures is actually the last historic value. Note that the collected data are raw values that need to be calibrated after adhering sensor and initializing it with the sensor reader, therefore a researcher that develops a non-official NFC-enabled app to read the sensor should calibrate it at least once with an accurate glucose reader (or compare the collected values with the ones actually shown by Abbot’s reader).Historic values: they are 32 values of 6 bytes that are located after the Trend values (from byte 125 to byte 316) and that represent the glucose concentration values obtained every 15 min for the last 8 h.Activity time: there are two bytes (317 and 318) that indicate the number of minutes since the sensor started to take measurements (i.e., since it was initialized with the official Abbot Freestyle reader), so they allow for obtaining the expiration date of the sensor.

#### 4.1.2. Decentralized Storage

In order to store the collected data, the proposed system makes use of the decentralized database OrbitDB [65], which runs over InterPlanetary File System (IPFS) [66]. Specifically, OrbitDB uses IPFS for storing data and for synchronizing multiple peer databases.

OrbitDB nodes can fetch content if there are connected peers in the swarm that share the database. Obviously, in order to replicate a database, at least one node must be connected to the swarm. Nodes can use the REST API to create/open a database and then add data to them. In the system implemented for this article the database was defined as an eventlog, which creates an append-only log. Fetching and synchronizing data are automatically orchestrated by OrbitDB, so this functionality does not need to be implemented by the REST API.

During the experiments presented in this article OrbitDB nodes were running on a fog gateway and in a virtual machine on a remote cloud on the Internet. In this latter case, the virtual machine make use of two AMD Opteron 3.1 Ghz cores and 8 GB of RAM.

#### 4.1.3. Smartphone App

The smartphone app was developed for Android, since, as of writing, nearly 80% of the market share belongs to such an operating system. Specifically, the smartphone used during the experiments for this article was an OnePlus 6T, which embeds an octa-core CPU (that runs at up to 2.8 Ghz), 6 GB of LPDDR RAM and 128 GB of internal memory.

Regarding the Android app, it was designed to provide a fast, easy and reliable way to retrieve patient’s data. Figure 4, on the left, shows a screenshot of the main screen of the app, where a speedometer widget points at the current patient blood glucose concentration values (the screenshot shows real values from one of the researchers that wore the system for 14 days). The main screen also shows a graph with the evolution of the patient glucose on a daily, weekly and monthly basis and a table with the minimum, maximum and average values during the selected time period.

The smartphone app also implements the first notification layer, which is triggered after reading the values collected through NFC from the sensor and when such values are not within the predefined thresholds.

To increase the reliability of the system, data are first stored locally on the smartphone via a SQLite database, which is queried from the Android app by using Room persistence library. Then, the collected data can be saved in a fog node (in OrbitDB) using a REST Application Programming Interface (API) [67]. When the app detects that there is not a connection with a fog gateway, the REST API requests are sent to another OrbitDB nodes on the Internet (during the tests performed in this article, such nodes were in a remote cloud).

#### 4.1.4. Back-End and Front-End

A web interface was designed and implemented to show historical values (i.e., a graph of the glucose levels) in order to ease medical supervision. Moreover, notifications are sent to the medical services whenever anomalous values of glucose are detected.

The required systems are deployed in a cloud machine that is also running an OrbitDB node and a service that exposes a REST API that obtains the data from the database, thus allowing other systems to access the decentralized data in a traditional way.

The visualization panel consists of a web interface designed as a Single Site Application using the Vue.js framework [68]. The application loads into the user’s web browser and makes HTTP requests to the REST API to show patients’ glucose historical data.

Along with the web application (showed in Figure 4, on the right) it is executed an event notification service. Such a service is triggered every time a user accesses the visualization interface and processes the data, thus detecting whether the glucose values are or not within certain pre-established thresholds. If the collected value is above or below such thresholds, an alert is generated and an SMS or a Telegram message [69] is sent to the medical services to warn them. For example, such messages can serve as a reminder for a nurse that a specific patient has an incoming hypoglycemia. Specifically, the notification service uses Twilio’s API [70] to send SMS, while a Telegram bot was developed to warn remote users through Telegram messages. As an example, Figure 5 shows SMS and Telegram messages sent to notify hypoglycemia and hyperglycemia alerts.

#### 4.1.5. Distributed Ledger

Ethereum was chosen as blockchain since the distributed ledger required by the proposed architecture has to be able to run smart contracts. Although the developed system can be deployed in the public Ethereum blockchain, for the experiments performed in this article, it was run on two testnets (i.e., Ethereum test networks): Rinkeby [71] and Ropsten [72]. Rinkeby and Ropsten differ in their consensus algorithm: while Rinkeby uses Proof-of-Authority (PoA) (with Clique PoA as consensus protocol), Ropsten uses Proof-of-Work (PoW). This means that, in the case of Rinkeby, there is a subset of authorized signers that assume block minting and Ether cannot be mined (it is requested through a faucet [73]). In contrast, in Ropsten Ether is mined, although it can be also requested from a faucet [74]. In practice, Rinkeby is able to create blocks on an average of 15 s, while Ropsten usually requires up to 30 s.

Smart contracts can be executed both in Rinkeby and Ropsten. For this article, the smart contracts were compiled and deployed by using Truffle [75]. The main smart contract used by the proposed system was developed to implement the meta-coin incentive mechanism detailed in Section 3.2, which initializes user account balance for the participating parties and manages the performed GlucoCoin payments. User wallets increase their balance with a fee when the owner of the wallet sends data to OrbitDB. If the user needs to buy a new glucose sensor, the account balance is decreased. The mechanism that performs the sensor purchase is implemented outside the blockchain (i.e., it is handled by an external provider). Since the expiration information is stored in OrbitDB, it can be used to trigger automatically the purchase and payment of the new sensor.

#### 4.1.6. Fog Gateway

The fog gateway of the cture may be implemented on most current SBCs, like Raspberry Pi 3 [76], Banan Pi [77] or Odroid-XU4 [78]. Among such devices it was selected the Orange Pi Zero Plus [79], since it provides a good trade-off between features and cost (it can be currently purchased for less than $20). The Orange Pi Zero Plus features an Allwinner H2 Sstem on Chip (SoC) that embeds a quad-core ARM Cortex-A7 microcontroller. The selected SBC also includes 256 MB of DDR RAM and includes interfaces for USB 2.0, 10/100 M Ethernet and WiFi (IEEE 802.11 b/g/n). In addition, it is worth noting that the SBC is really small (48 mm × 46 mm) and lightweight (26 g).

The fog gateway runs on ARMbian [80] and includes the following main software to implement the proposed architecture:OrbitDB and IPFS. Every fog gateway is actually an autonomous OrbitDB node that runs on IPFS and that synchronizes periodically with the other OrbitDB nodes deployed on the Internet.Web3 [81]. It is required to interact with the Ethereum blockchain. Specifically, Web3 is used as a JavaScript API to exchange requests with Infura [82], which provides an easy-to-use HTTP API that allows for accessing the Ethereum blockchain even from resource-constraint IoT devices.Node.js [83]. It is needed for executing JavaScript code outside of a browser, thus easing the interaction with Web3 and OrbitDB. In the case of Web3, in practice, the proposed implementation runs a Node.js instance that performs calls to the Web3 API in order to exchange requests with Infura, which interacts with Ethereum.

Finally, it is worth pointing out that every fog gateway makes use of two WiFi interfaces simultaneously. One of them is the embedded interface, while an external WiFi adapter is plugged into the Orange Pi Zero Plus to provide the second one. The reason for the use of both interfaces is the need for connecting wirelessly to the Internet through one interface, while the second one is setup in Access Point mode to broadcast a Service Set IDentifier (SSID) and a MAC address that can be identified easily by the smartphones of patients in order to connect to them fast.

### 4.2. Regular Functioning of the System

Three common use cases may arise during the regular operation of the system. The first one is illustrated in Figure 6 and represents the insertion process that is performed with the collected data. It must be noted that, before starting to use the system, it needs to be initialized: the decentralized storage nodes require to be synchronized through IPFS (steps 0A and 0B in Figure 6), the used smart contracts need to be compiled and deployed in a Ethereum testnet (step 0C), the GlucoCoin wallets have to be initialized (step 0D) and the CGM sensor has to be placed on the patient’s arm. Moreover, the developed mobile app has to be installed on the patient’s smartphone and the required Android permissions need to be granted so that the app can make use of the WiFi interface to detect and connect to the available fog gateways.

Once the system is deployed and configured, the patient can read the sensor by approaching the smartphone (step 1) (note that there are amateur [63,64] and commercial [84] devices that make use of arm bands to automate this reading process). If there is a fog gateway in range, the collected glucose values are sent to it, which stores them in OrbitDB (step 2A). If there are no fog gateways in range, the smartphone sends the data to an OrbitDB node on the Internet (step 2B), which appends them to its database (step 3). Since the stored data are decentralized and therefore stored on synchronized OrbitDB nodes, an external authorized provider can access such data (step 4) and reward the patient with a number of GlucoCoins that is proportionate to his/her data contribution (step 5).

A second relevant use case is illustrated in Figure 7 and is related to the notifications performed by the system when a dangerous situation is detected (e.g., when detecting an incoming hypoglycemia). In such a case, after initializing the system (steps 0A and 0B), glucose concentration values can be read (step 1) and three notification layers may warn the user:Local warnings. For simple analyses, a smartphone is powerful enough to process the collected data and warn the user about a dangerous situation. Since the sensor data can be stored in the local database, the smartphone app can read such an information and then show notifications to the user (step 2).Fog warnings. If there is a fog gateway in range, the smartphone will send the collected values to it (step 3A) in order to store them in OrbitDB. The collected data can be accessed by the internal mHealth fog service, which can warn the patient (step 4A) and the local users (step 5A) in case of detecting a dangerous situation.Cloud warnings. If the smartphone detects no fog gateways, it sends the collected data to another OrbitDB node on the Internet (step 3B), which stores them (step 4B). Such data are retrieved periodically by the back-end (step 5B) in order to essentially show them to remote users through a web interface. In addition, the back-end processes the collected data (step 6B) and, if a dangerous situation is detected, it can warn local and remote users (steps 7B and 8B).

The third relevant use case is depicted in Figure 8 and illustrates an example of smart contract execution. Specifically, this third use case refers to how a new sensor can be purchased automatically by relying on the data provided by the CGM sensor. Such a process begins after the initialization of the system (steps 0A, 0B, 0C and 0D) and requires reading the expiration date (actually, the activity time) of the sensor (step 1) and storing it either on the local fog gateway (step 2A) or on a remote OrbitDB node (steps 2B and 3). The stored data are monitored by an oracle (step 4) that feeds a smart contract (step 5) that decides whether it is time to perform the purchase of a new sensor. When such a time comes, the smart contract is executed: it is checked the patient’s GlucoCoin wallet (step 6) and, if there are enough GlucoCoins, the purchase is performed (step 7). Once the sensor provider confirms the purchase (step 8), the price of the new sensor in GlucoCoins can be withdrawn from the user wallet (step 9) and the provider can send (for instance, by courier) the CGM sensor to the patient (step 10).

## 5. Experiments

In order to evaluate the proposed architecture, several experiments were designed to provide insightful results about three different aspects: (1) the throughput of the fog and cloud architecture, (2) the performance of the decentralized database, and (3) the performance of the blockchain.

### 5.1. Baseline Performance of Fog and Cloud Nodes

Before studying the performance of the proposed architecture in detail, a first preliminary test was carried out for establishing a performance baseline for the fog and cloud nodes, since their constrained hardware (in the case of fog nodes) and remote connectivity (in the case of the cloud nodes) may limit the overall performance of the proposed system. Thus, an Orange Pi Zero Plus and a cloud node were evaluated when only running a Node.js server. Specifically, each test performed 5000 connections on a fog node and 1000 on a remote node (to avoid network congestion issues) at different connection rates in order to determine the node maximum throughput up to the point where connection errors arise.

The obtained results are shown in Figure 9, where it is represented the desired request rate (the one imposed by the tests) versus the rate actually achieved during the tests. As it can be observed in the Figure, the desired and real rates are roughly the same up to a point when the node is not able to handle the requests and thus its performance decreases. Specifically, the fog node reaches its peak performance at 300 requests per second, while this point is at 200 requests per second for the cloud node. In the case of the fog node, this is due to its hardware constraints, while, in the case of the cloud node, it is related to the restrictions of its network (i.e., the load of the network and the characteristics of the devices involved in processing and routing the requests through the Internet). In any case, these results are a useful reference when evaluating the performance analyses described in the next subsections.

### 5.2. Performance of the Decentralized Database

In order to estimate the throughput of the selected decentralized database, OrbitDB nodes were deployed locally (in fog nodes) and remotely (in a cloud). In such scenarios it was measured the average time required by an OrbitDB node for processing each REST API request. For the sake of fairness, tests were performed for four different payload sizes to evaluate their effect on network delay (for each payload size, the time required for processing 1000 requests was averaged).

The obtained results are shown in Figure 10 and Figure 11. As it can be expected, the larger the payload, the larger the time response and the lower the request rate. Moreover, it can be observed that the cloud-based OrbitDB node is clearly slower in spite of being more powerful than the fog node. In the worst evaluated case (i.e., for the cloud and a 4 KB payload), the decentralized node, although it is only able to handle two requests per second, it is actually really fast, since it is able to process and to respond to each of the requests in less than half of a second, which seems to be quick enough for most glucose monitoring applications.

It is also worth noting that the difference in time response between the fog node and the cloud increases as payload size gets larger, mainly due to the communications transmission time through the network (i.e., although the processing time required by the OrbitDB node remains constant, the time required to exchange the request payload increases). In fact, during the performed experiments, the average round trip time (calculated using the ’ping’ command in Linux, which makes use of 56-byte packets) for the fog node was 0.859 ms, while the same for the cloud was 45.920 ms, which makes a significant difference.

Figure 12 and Figure 13 show the results of a second test, which measured the performance of the decentralized database when carrying out 2000 consecutive write operations on an OrbitDB node that ran on a fog and on a cloud node. For the sake of fairness, the data of the Figures were obtained by using the official OrbitDB benchmarking scripts [85], which obtain the average throughput of the OrbitDB node every 10 s.

While in Figure 12 it can be observed that the fog OrbitDB node throughput oscillates between 3.7 and 4.5 write requests per second, the cloud node throughput shown in Figure 13 can reach between 3.5 and 6 requests per second. This means that the fog node responds faster than the cloud node, but its hardware is less powerful than the one used by the cloud, so it is not able to process as many requests per second. However, it must be noted that both in the fog and in the cloud scenarios the average throughput oscillates continuously due to different factors, like the computational and network load (i.e., the network is actually shared with other users).

A third test was carried out in order to determine the performance of an OrbitDB node when carrying out two main operations: when fetching content from connected peers in the swarm and when replicating content in other connected peers (these both operations are automatically orchestrated by OrbitDB). Figure 14 shows the average throughput for every 10 s when performing 2000 fetching and replication operations (from a fog node to a cloud node). Despite the observed oscillations, it can be concluded that the average fetching throughput is slower than the one related to a replication due to the complexity of this latter operation.

### 5.3. Time Response of Event Warnings: Local, Fog and Cloud

As it was previously indicated in Section 4.2, when a dangerous situation for the patient is detected, event notifications are issued. Depending on the target user, it can be distinguished among local, fog and cloud warnings. In order to measure the delay of these warnings, it can be averaged a number of significant requests (i.e., 500 requests for the fog node, 100 for the cloud and 20 for Android) in the three aforementioned scenarios.

In the first column of Figure 15 it is represented the average time required by the tested Android smartphone app to read the information from the local database and show a warning to the patient. As it can be observed, such a time is really small: only 11.9 ms.

The second column of Figure 15 represents the average notification time of a fog node when sending a warning. Such a time is exactly 249.5 ms, and includes the time required to send a request, to process it (thus interacting with OrbitDB) and to receive the notification from the fog gateway. The third column of Figure 15 shows the longest average notification time, which is obtained when performing the same tasks previously indicated for the fog gateway, but for an OrbitDB node that ran on the cloud. In this case, the average response time was 465.8 ms.

These results for the three selected scenarios may be used when developing applications that have to comply with strict latency requirements related to critical reaction times that might be needed to warn or to send certain information to DM patients or to health devices carried by them.

### 5.4. Blockchain Performance: Smart Contracts Execution Time

In order to detect potential bottlenecks in the proposed architecture, an additional experiment was performed to measure the response latency of the blockchain. This is specially interesting in the case of using Ropsten testnet, where the time to mint a block may vary noticeably as it uses PoW as consensus protocol.

Figure 16 shows the response latency of 1000 transactions in the Ropsten testnet during the execution of the smart contract related to the management of GlucoCoin (its main code is shown in Listing 1).

In Figure 16 it can be observed that the smart contract execution time oscillates continuously mainly due to the number of available miners (i.e., participants that execute the smart contract) and their computational load. Specifically, the obtained blockchain response times vary significantly from less than 20 s up to more than 240 s. Nonetheless, the average blockchain time response is 36.47 s, which is not very different from the usual average time provided by Ropsten (roughly 30 s), although the standard deviation of the performed measurements was 33.96 s. These response times can be considered normal in a blockchain based on a PoW consensus mechanism, since it includes the time for sending the transaction request, the time required by the consensus protocol to select the miner that will execute the smart contract and the time needed for executing the smart contract. As a reference, it can be indicated that, in a PoW-based cryptocurrency like Bitcoin [86], the transaction execution time in June 2019 was 9.47 min [87]. In any case, it is worth pointing out that execution delay obtained in Ropsten is noticeably larger than the time response required by OrbitDB nodes, so it will have to be taken into consideration in cases where response latency is essential.

It is worth pointing out that, thanks to the use of a blockchain, any reader can check the obtained results by browsing the wallet used for the experiments on the Ropsten testnet [88]. In addition, Figure 17 shows a screenshot of the Ethereum wallet that includes the details of some of the transactions performed on the testnet.

Listing 1: Main code of the smart contract that controls GlucoCoin.

		pragma solidity >=0.4.25 <0.6.0;

		contract GlucoCoin {
		    mapping (address => uint) balances;

		    event Transfer(address indexed _from address indexed _to, uint256 _value);

		    constructor() public {
		            balances[tx.origin] = 10000;

		    }

		    function sendCoin(address receiver, uint amount) public returns(bool sufficient) {
		            if (balances[msg.sender] < amount) return false;
		            balances[msg.sender] −= amount;
		            balances[receiver] += amount;
		            emit Transfer(msg.sender, receiver, amount);
		            return true;
		    }

		    function getBalanceInEth(address addr) public view returns(uint) {
		            return ConvertLib.convert(getBalance(addr),2);
		    }

		    function getBalance(address addr) public view returns(uint) {
		            return balances[addr];
		    }
	      }



## 6. Future Work

Although the tests performed on the proposed system showed its practical feasibility, there are several aspects that can be further improved:Automatic readings. Currently, patients have to approach their NFC-enabled smartphone to the CGM sensor to read it. Future work should be performed to design and implement solutions that automate such a process or to adapt already available amateur [63,64] and commercial [84] devices.Optimization of the hardware and software baseline performance. Although the obtained amounts of requests per second are high enough for the fog gateway (300 requests per second), the cloud should be improved in order to go above the achieved 200 requests per second.It should be further analyzed possible enhancements on the decentralized database performance, since in the experiments it was only possible to reach up to 6 write requests per second, which can be too restrictive in some practical scenarios.The response time of the blockchain can be improved by using faster consensus mechanisms or other new improvements for Ethereum, like sharding [89], Raiden [90] or Plasma [91].It would be ideal to perform trials on large sets of population and analyze the real performance of the system, the behavior of the patients regarding the incentive mechanisms and the usefulness of the collected data.

## 7. Conclusions

This article presented the design, implementation and evaluation of an IoT CGM-based system for mobile crowdsourcing diabetes research and care. Such a system is able to collect blood glucose levels from CGMs that can be accessed remotely. Thus, the system allows for monitoring patients and warn them in real-time in case a dangerous situation is detected. In order to create the proposed system, a fog computing system based on distributed mobile smartphones was devised to collect data from the CGMs and send them to a remote cloud and/or to a blockchain. Thanks to the blockchain and the proposed CGM-based system it is possible to provide a transparent and trustworthy blood glucose data source from a population in a rapid, flexible, scalable and low-cost way. Such mobile crowdsourced data can enable novel mHealth applications for diagnosis, patient monitoring or even public health actions that may help to advance in the control of the disease and raise global awareness on the increasing prevalence of diabetes. Furthermore, the performance of the proposed blockchain-based architecture, the decentralized database and the smart contracts were evaluated in diverse scenarios that take into account the latency demands that may be required by different potential stakeholders (e.g., patients, doctors, caretakers, scientists, government, insurance companies, governments, app developers) of the healthcare ecosystem.

## Figures and Tables

**Figure 1 sensors-19-03319-f001:**
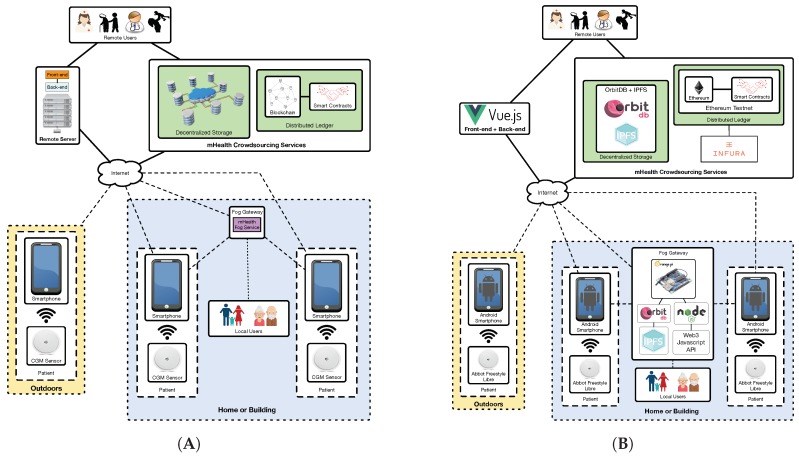
Proposed communications architecture (**A**), and implemented architecture (**B**).

**Figure 2 sensors-19-03319-f002:**
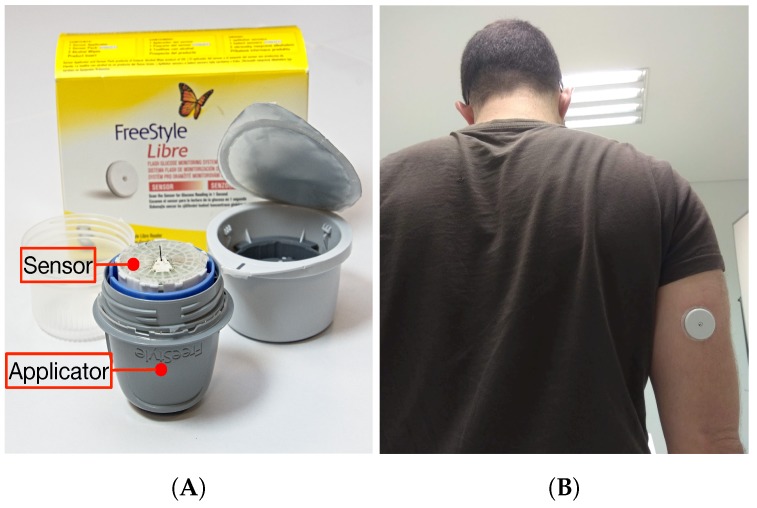
Abbott’s Freestyle Libre sensor and applicator (**A**), and adhered sensor (**B**).

**Figure 3 sensors-19-03319-f003:**
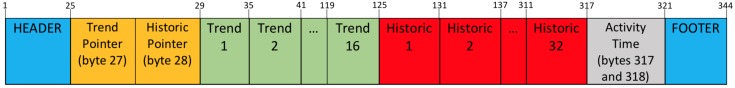
Structure of the internal memory (in byte) of Abbot’s Freestyle Libre sensor.

**Figure 4 sensors-19-03319-f004:**
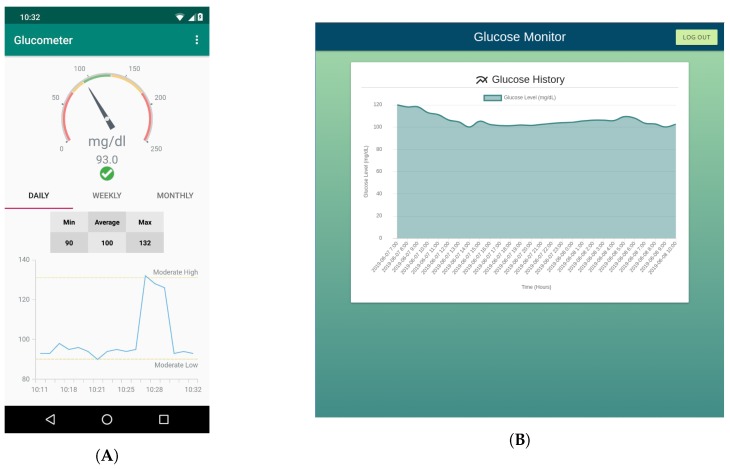
Screenshot of the smartphone app (**A**) and of the web application (**B**).

**Figure 5 sensors-19-03319-f005:**
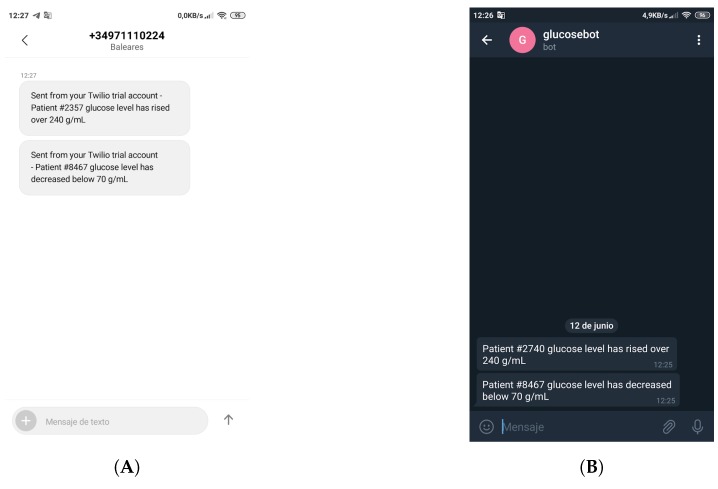
Event notifications received through a Twilio’s SMS (**A**) and a Telegram message (**B**).

**Figure 6 sensors-19-03319-f006:**
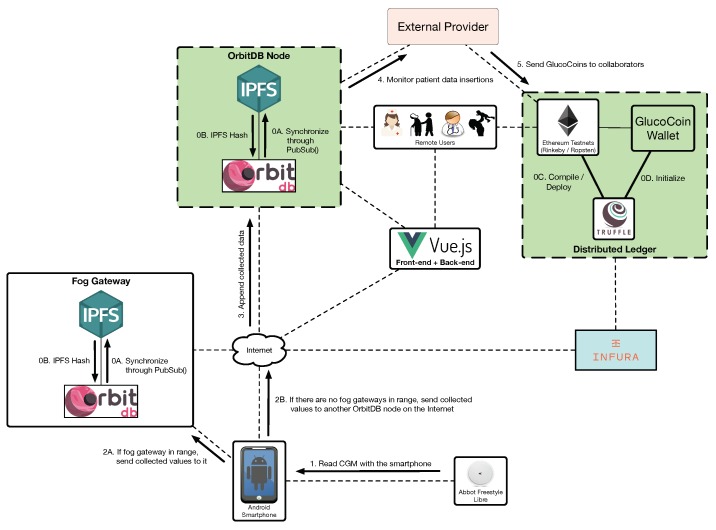
Insertion process for the data collected from the CGM.

**Figure 7 sensors-19-03319-f007:**
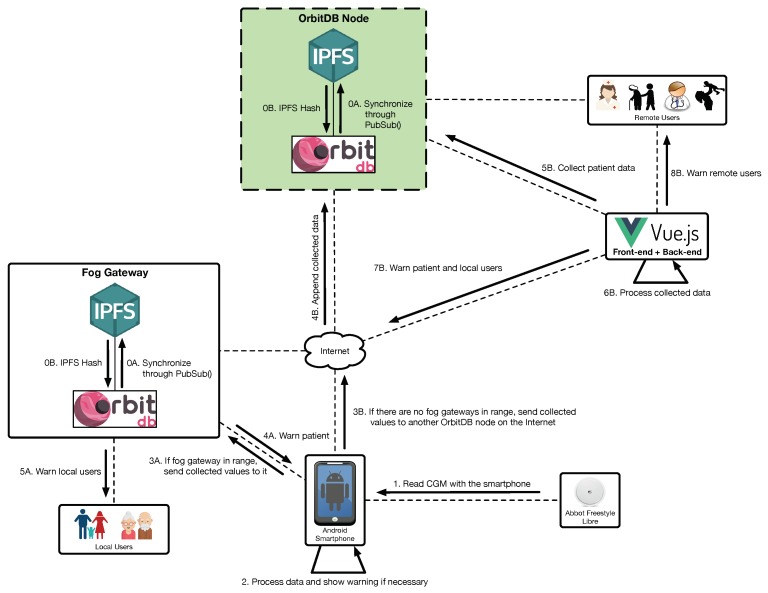
Patient and user notification processes.

**Figure 8 sensors-19-03319-f008:**
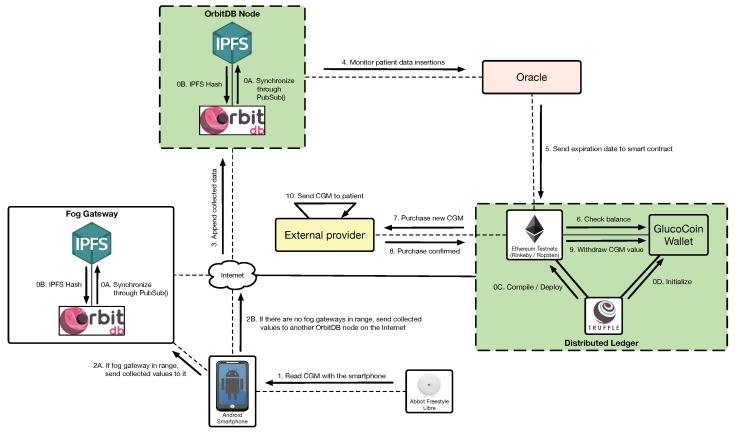
Process to automatically purchase CGMs due to the imminent expiration of the current one.

**Figure 9 sensors-19-03319-f009:**
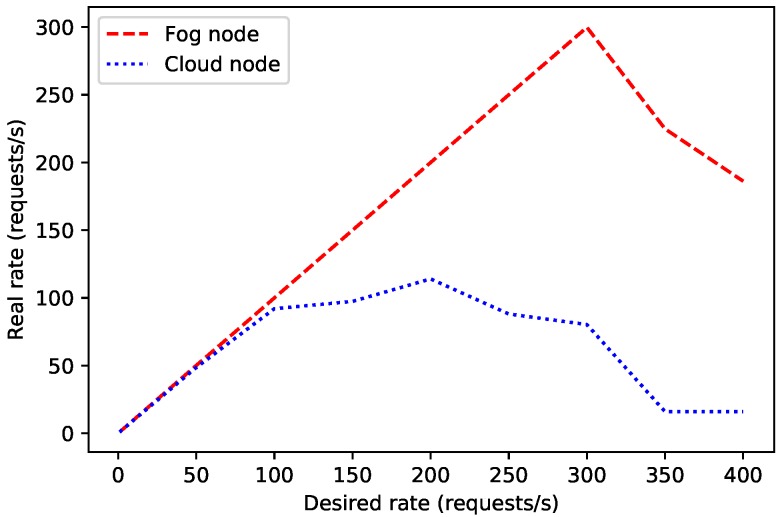
Desired/real request rate for 5000 connections on the fog node and 1000 on the cloud node.

**Figure 10 sensors-19-03319-f010:**
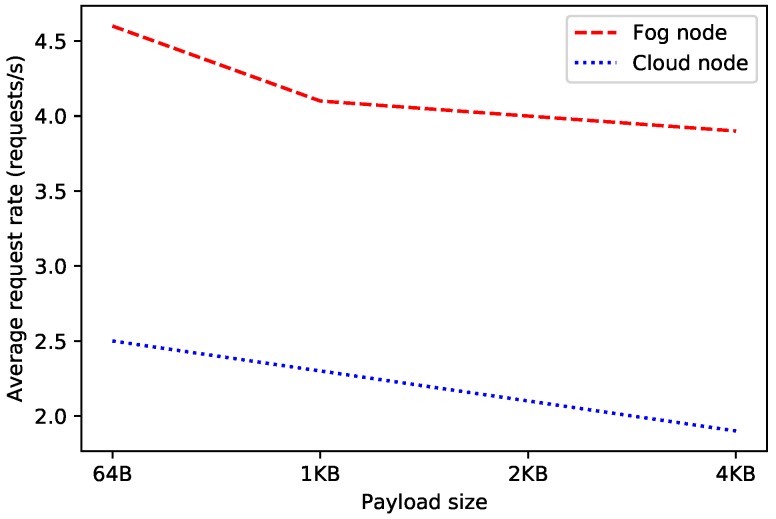
Average request rate for OrbitDB when running on fog and cloud nodes.

**Figure 11 sensors-19-03319-f011:**
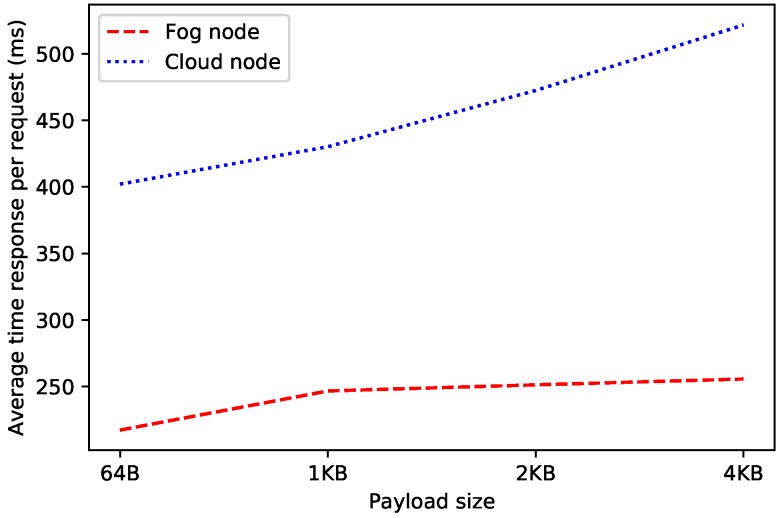
Average response time of OrbitDB when running it on fog and cloud nodes.

**Figure 12 sensors-19-03319-f012:**
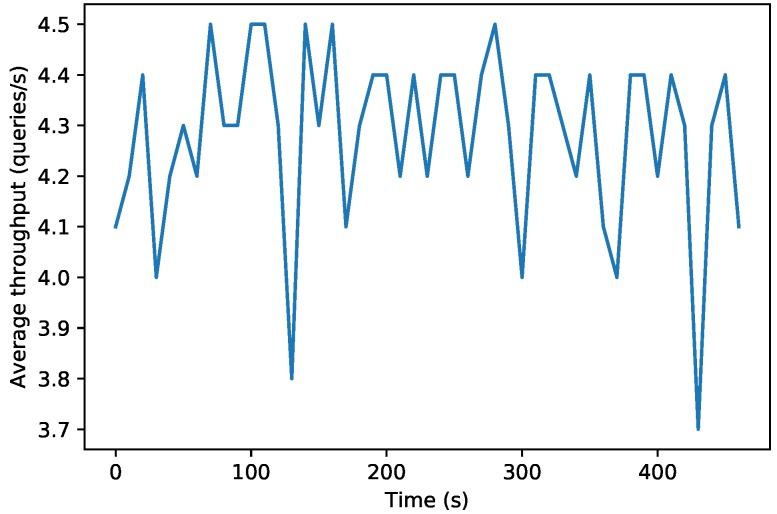
Performance of the fog OrbitDB node executing 2000 queries.

**Figure 13 sensors-19-03319-f013:**
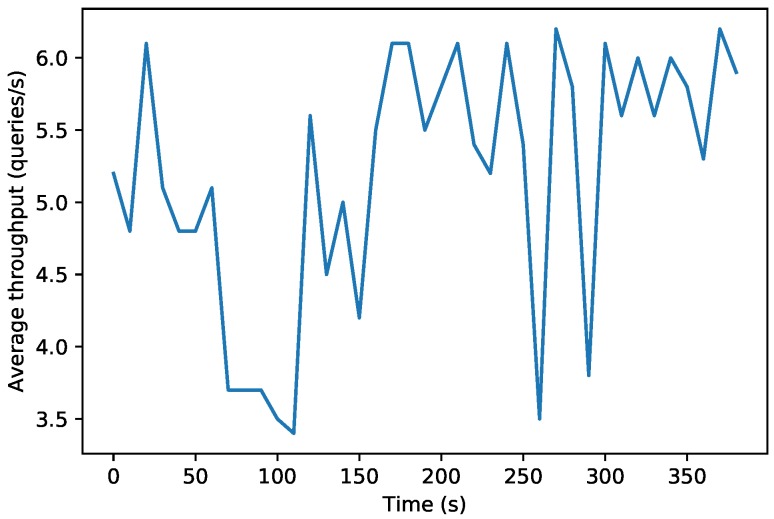
Performance of the cloud OrbitDB node executing 2000 queries.

**Figure 14 sensors-19-03319-f014:**
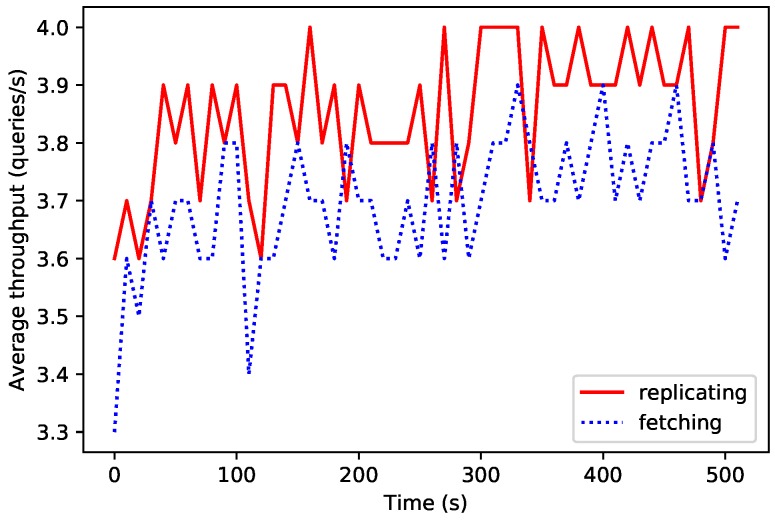
Performance in OrbitDB fetching and replication operations.

**Figure 15 sensors-19-03319-f015:**
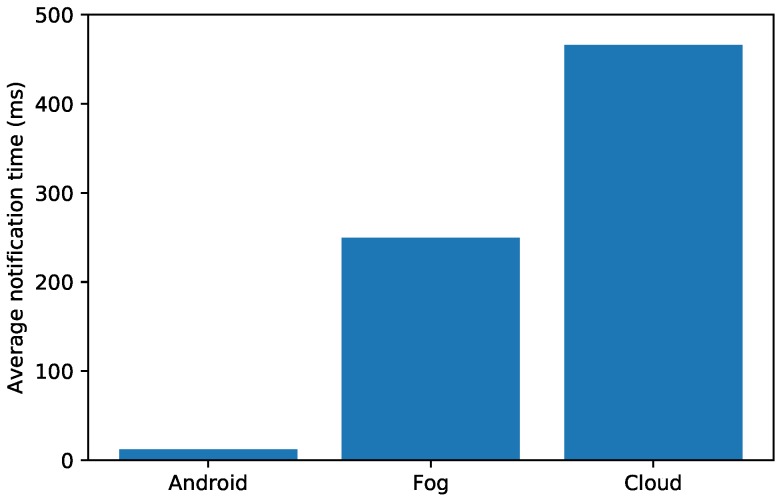
Average notification times in different scenarios.

**Figure 16 sensors-19-03319-f016:**
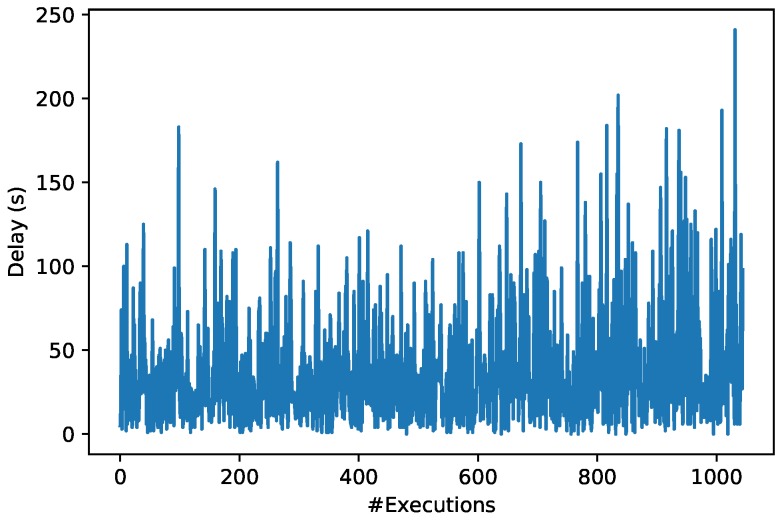
Ropsten testnet time response when executing smart contracts.

**Figure 17 sensors-19-03319-f017:**
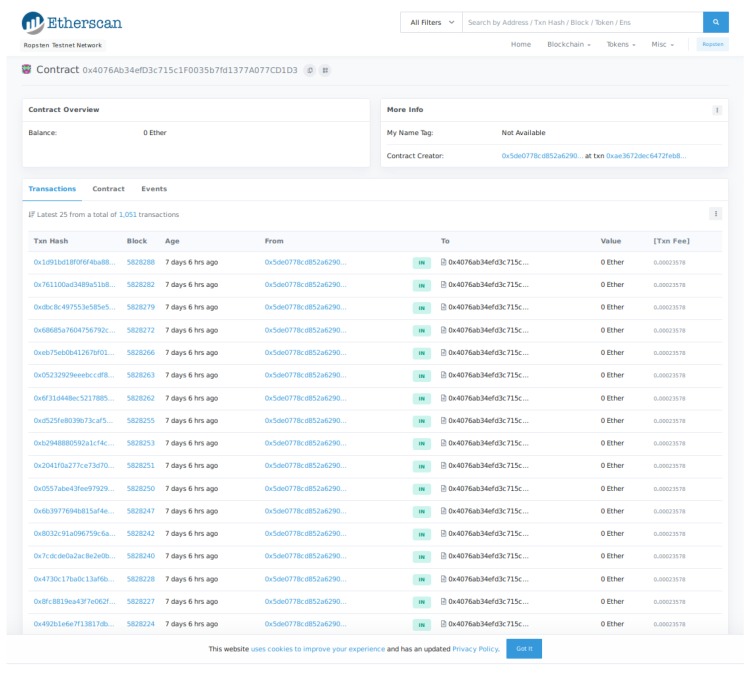
Ropsten testnet main view of the transactions related to the deployed smart contract.

**Table 1 sensors-19-03319-t001:** Comparison of the characteristics of the most popular CGMs.

	Models	Dexcom G4 Platinum	Dexcom G5 Mobile	Dexcom G6	Medtronic Guardian Connect	Medtronic iPro 2 Prof	Medtronic MiniMed 670G	Abbot Freestyle Libre	Abbot Freestyle Libre 2	Roche Eversense XL
Characteristics										
Duration		7 days	7 days	10 days	6 days	6 days	7 days	14 days	14 days	180 days
Daily calibration		Yes	Yes	No (manual calibration is available if the start-up calibration code is lost)	Yes	Yes	Yes	No	No	Yes
Alarms		Yes	Yes	Yes	Yes	No	Yes	No	Yes	Yes
Price		€ 1100 (Kit with a receiver, a transmitter and 4 sensors)	€ 1270 (Kit with a receiver, a transmitter and 4 sensors)	€ 1000 (Kit with a receiver, a transmitter and 3 sensors lasting 10 days each)	It varies depending on its use (ad-hoc, partial or intensive)	Not publicly available	Transmitter: $ 699, sensors ranging from $ 50–$ 75, depending on the number of purchased units	€ 169.90 (Kit with one reader and two sensors)	€ 169.90 (Kit with one reader and two sensors)	€ 399 for the transmitter and 1399 for one sensor
Data transmission		Automatic, every 5 min	Automatic, every 5 min	Automatic, every 5 min	Automatic, every 5 min	Automatic, every 5 min	Automatic, every 5 min	Manual	Manual	Automatic, every 5 min
Reading range		Max. 6 m	Max. 6 m	Max. 6 m	Max. 6 m	-	-	Max. 10 cm	-	
Remote monitoring		No	Yes (iOS/Android app)	Yes (iOS/Android app)	Yes (iOS app)	Yes (iOS/Android app)	-	Yes (Android app)	-	Yes (iOS app)
Communications		Bluetooth	Bluetooth	Bluetooth	Bluetooth	-	-	NFC	NFC, BLE	Bluetooth
MARD		13%	9%	9%	13.60%	11 %	8.7%	11.40%	9.5%	11.60%

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
