# Peer review of "Enabling the Internet of Mobile Crowdsourcing Health Things: A Mobile Fog Computing, Blockchain and IoT Based Continuous Glucose Monitoring System for Diabetes Mellitus Research and Care"

_sensors, 2019, doi:10.3390/s19153319_

Round 1
Reviewer 1 Report
The paper presents a proposal of an IoT architecture for diabetes research and healthcare. The system would include connection of specific sensors, fog computing and smart contracts thanks to blockchain. However, it is difficult to know what specific type of paper it is. On the one hand, the introduction is very long. In addition, it includes a section, related work, which includes a review of CGM, applications and manufacturers, fog computing and blockchain for health applications and Mobile Crowdsourcing Healthcare Services. All this ends up occupying more than 6 pages. All this seems like a part of a review paper instead a short introduction to present the main contribution.
The design of the architecture and, above all, its implementation also occupies a good part of the article. Too much space and too much detail is devoted to a single sensor, Abbott's Freestyle Libre. Finally, some experiments are described to evaluate the performance of different components.
On the other hand, although authors claim that an IoT architecture for healthcare is proposed, there is no real implementation and demonstration of the complete system. A single IoT device has been incorporated and no tests have been performed, even on a small scale, on the use of the complete system for healthcare.
In summary, to be accepted, the paper should be greatly reduced and be much more focused on actual contributions.
Author Response
Dear Sir/Madam,
The authors would like to thank the reviewer for his/her valuable comments, which have
certainly helped us to improve the manuscript. Please find attached our detailed responses to the
comments. In order to ease the labour of the reviewers we have colored in red the major
differences with the previous version of the article.
Best regards,
The authors.

Reviewer 2 Report
This paper describes in details the development of a new infrastructure for diabetes reasearch and care based on state-of-the-art technologies and continuous glucose monitoring. It was really interesting reading about how the authors built the whole system. The paper defines each aspect of the infrastructure in the very details providing every information needed to understand it. However, I found it really hard to read all the manuscript and follow the discussion through the last page, since the paper is very long. It actually looks more like a book/manual than a journal article.
My advice is to accept the manuscript after it is condensed in no more than 20 pages (rather than 31). Indeed, I think this will ease the reading and enhance the paper impact.
Author Response

(The authors gave the same response as above.)

Reviewer 3 Report
There is a need for someone who has English as there first language to review the text as some basic grammar errors do distract the reader from appreciating the underlying messages.
The paper is far too long and could be cut significantly without loosing any of the key points, The majority of the paper was taken up by an overly long description of diabetes, with some of the statements made being sweeping and lack clinical accuracy, This section could be cut significantly as it adds very little to the key purpose of the paper. Like wise the following sections can also be significantly cut back. I personally found that the writing style was more in keeping with a book chapter than a journal article so would suggest the authors seek further local independent academic peer review on content and style prior to resubmission.
That being said. The topic is of interest and the key points, when you get to them in the paper are indeed relevant.
Author Response

(The authors gave the same response as above.)

Reviewer 4 Report
This article shows the design and implementation of a system that enhances commercial CGMs by adding IoT capabilities to them that allow for monitoring patients remotely and thus warning them in dangerous situations.The topic is interesting and it has a good contribution. The authors should improve some aspects as: avoid to use enough bullets in the manuscript, add future work and explain more Figure 2 and Figure 3. Reading all sections, the explanations are coherent and are well-argued. In page 24, the authors should explain more the following: "As it can be observed, the blockchain response time varies significantly from less than 20 s up to more than 240 s (the average is 36.47 s and the standard deviation, 33.96 s). In addition, it is worth pointing out that execution delay is noticeably larger than the time response of OrbitDB nodes, so it will have to be taken into consideration in cases where response latency is essential". Why do you think these results are good?
Author Response

(The authors gave the same response as above.)

Round 2
Reviewer 1 Report
Based on the changes made the quality of the manuscript has been improved, especially with regards to the implementation, experiments, and future work. However, there are still minor changes that need to be made.
1. Fingerpricking is mentioned several times in the abstract. Although it is correct, it is more appropriately called self-monitoring of blood glucose (SMBG). I suggest to reword the section of the abstract to reduce the amount of times pricking is mentioned as it should only be mentioned once.
For more information on how to correctly describe SMBG please refer to: Self-Monitoring of Blood Glucose: The Basics. Evan M. Benjamin, MD, FACP. Clinical Diabetes 2002 Jan; 20(1): 45-47. https://doi.org/10.2337/diaclin.20.1.45
2. “Blood glucose” should be used instead of “blood sugar” in all instances
3. Figure 1 and 2 are essentially the same. In fact, figures 1, 2, 7, and 8 should be combined and/or compacted to contain the same information but occupy less space.
4. Line 2: Not “or” but “and”. These can be affected simultaneously
5. Line 10: “send” should be changed to “sends”
6. Line 25: change to: which is usually referred to as Diabetes
7. Line 27: Diabetes leads to both micro and microvasculature changes, which affect the kidneys (nephropathy), nerves (neuropathy), cardiovascular system, and eyes (retinopathy). Please explain these complications in a concise and comprehensive manner. The comment in brackets is confusing and does not give any explanation of changes that incur blindness.
8. Line 29: Type 1 is an autoimmune process, which leads to grossly diminished insulin levels. Type 2 – arises from compromised function of insulin-producing cells in the pancreas and insulin resistance in the peripheral tissues.
9. Lines 29-31. I would suggest listing the 3 types and then providing a brief (maximum one sentence) explanation of each type instead of putting the explanation in brackets.
10. Line 37: Diabetics is not commonly used. I suggest to use “DM patients” or “those with diabetes”
11. Line 38: change to: glucose, which allows patients the ability to make informed therapeutic decisions.
12. Line 39: glucose highs and lows should be changed. Generally hyperglycemia (high blood glucose levels) and hypoglycemia (low blood glucose levels) are used.
13. Line 42: this delay is much greater
14. Line 46: change to: and are able to make autonomous decisions when actions must be performed as fast as possible to avoid dire outcomes.
15. Line 48: change to: where rule-based decisions can be taken
16. Line 51: Instead of “its massive use”, would “overloading” be more appropriate here?
17. Line 52: change to: due to physical distance, there may be a lag between the cloud and the patient resulting in a delay between the decision to perform an action and the communication to the patient to do so, which may be too long in some cases.
18. Line 60: change “arises” to “arise”
19. Line 70: change “on massive sets of population” to “on a large scale”
20. Line 70: change to: is one potential technique
21. Line 76: too many commas
22. Lines 81-92: You must decide if you want the bullet points to be in past tense or present tense (below I have made changes to the present tense but they can be change to past tense if it is preferred by the authors)
23. Line 81: bullet 1 : A description of an IoT CGM-based system able to monitor glucose concentration values remotely and quickly inform patients and/or their care givers of a dangerous
24. Line 83: bullet 2: The design of a system able to provide fast warnings to using fog computing gateways when the user is located in specific scenarios (e.g. inside the patients home, in a hospital or in a nursing home)
25. Line 87: bullet 3: A description of a mHealth blockchain-based decentralized architecture that does not depend on any central authority
26. Line 89: bullet 4: The proposal of a data crowdsourcing mechanism that encourages collaboration by rewarding patients.
27. Line 91: bullet 5: The evaluation of the proposed architecture to assess the performance of the described decentralized storage and blockchain
28. Line 110: change to: A CGM is also able to decipher the influence of daily habits on the health of DM patients
29. Line 116: change to: insulin, with physical activity being one of the most challenging as it may result in hypoglycemia in Type-1 DM patients
30. Lines 114-122: I do not think that this paragraph is necessary
31. Line 124: change “has achieved a remarkable success” to “has achieved remarkable success”
32. Line 129: change to: which derives several advantages:
33. Line 167: change “As of writing” to “Currently”
34. Line 176: add comma after crowdsourcing and remove comma after essential
35. Line 215: remove the word “basically”
36. Line 219: replace “what” with “which”
37. Line 240: start a new sentence: The rest of the processes (e.g. ….)
38. Line 241: What is the meaning of transparent in this context? Do you mean that the user is not responsible for these actions, and that these actions are automated? Please provide further explanation here
39. Lines 301-305: Was this performed in this implementation? If so, this should be described.
40. Line 405: change to: The reason for the use of both…
41. Line 495: change “what” to “which”
Author Response
Dear Sir/Madam,
The authors would like to thank again the reviewer for his/her meticulous review and very useful comments, which have certainly helped us to improve the manuscript. Please find below our detailed responses to the comments. In order to ease the labour of the reviewers we have colored in red the major differences with the previous version of the article.
Best regards,
The authors.

Reviewer 3 Report
The paper reads much better now and presents a more coherent description of the work with relevant supporting information.
Author Response
Dear Sir/Madam,
The authors would like to thank again the reviewer for his/her meticulous review and very useful comments, which have certainly helped us to improve the manuscript.
Best regards,
The authors.
Reviewer 4 Report
This paper can be accepted in the current form.
Author Response

(The authors gave the same response as above.)
